# Validation of an Automated Optical Scanner for a Comprehensive Anthropometric Analysis of the Foot and Ankle

**DOI:** 10.3390/bioengineering10080968

**Published:** 2023-08-16

**Authors:** Riccardo Sacco, Marie-Aude Munoz, Fabien Billuart, Matthieu Lalevée, Julien Beldame

**Affiliations:** 1Department of Orthopedic Surgery, Rouen University Hospital, 37 Bd Gambetta, 76000 Rouen, France; riccardo.sacco@chu-rouen.fr (R.S.); matthieu.lalevee@chu-rouen.fr (M.L.); 2Centre Médical Achille, 200 Avenue des Prés d’Arènes, 34070 Montpellier, France; docteur.munoz@gmail.com; 3UFR Simone Veil-Santé, Université de Paris-Saclay, 2, Avenue de la Source de la Bièvre, 78180 Montigny-le-Bretonneux, France; 4Laboratoire D’analyse du Mouvement, Institut de Formation en Masso-Kinésithérapie Saint-Michel, 68, Rue du Commerce, 75015 Paris, France; 5Unité de Recherche ERPHAN, UR 20201, UVSQ, 78000 Versailles, France; 6ICP-Clinique Blomet, 136 Bis Rue Blomet, 75015 Paris, France; 7Clinique Megival, 1328 Avenue de la Maison Blanche, 76550 Saint Aubin sur Scie, France

**Keywords:** foot and ankle measurements, laser scanner, anthropometric data, reproducibility

## Abstract

**Background:** Our objective was to conduct a comprehensive analysis of the reproducibility of foot and ankle anthropometric measurements with a three-dimensional (3D) optical scanner. **Methods:** We evaluated thirty-nine different anthropometric parameters obtained with a 3D Laser UPOD-S Full-Foot Scanner in a healthy population of twenty subjects. We determined the variance of the measurements for each foot/ankle, and the average variance among different subjects. **Results:** For 40 feet and ankles (15 women and 5 men; mean age 35.62 +/− 9.54 years, range 9–75 years), the average variance was 1.4 ± 2 (range 0.1 to 8). Overall, the mean absolute measurement error was <1 mm, with a maximum variance percentage of 8.3%. Forefoot and midfoot circumferences had a low variance <2.5, with variance percentages <1%. Hindfoot circumferences, malleolar heights, and the length of the first and fifth metatarsal to the ground contact points showed the highest variance (range 1 to 7). **Conclusions**: The UPOD-S Full-Foot optical Scanner achieved a good reproducibility in a large set of foot and ankle anthropometric measurements. It is a valuable tool for clinical and research purposes.

## 1. Introduction

As technology continues to advance, three-dimensional (3D) scanners have increasingly replaced conventional methods such as ink prints and tape measure for evaluating foot and ankle anthropometrics [1,2]. They offer fast acquisition speed, user-friendly features, and improved accuracy compared to conventional methods [3,4,5,6,7]. Moreover, 3D scanners provide the possibility to measure several anthropometric parameters, which would be highly time consuming with standard methods [4,8,9,10,11].

The clinical application of anthropometric measures of the foot extends to various fields, including orthopedics, podiatry, sports medicine, and footwear design [9,12,13,14,15]. Anthropometric measures of the foot and ankle have demonstrated clinical utility across various domains of healthcare [16,17]. They offer valuable insights into foot morphology, aiding in the diagnosis, treatment, and prevention of foot-related conditions. In the last few years, there has been an increasing body of scientific literature disseminating research on foot anthropometry and contributing to advancements in the field, with potentially important clinical implications [16,18,19,20]. The precision and accessibility of foot anthropometry are expected to further improve, enhancing its clinical utility in the years to come [17].

In orthopedics, these measures assist in the diagnosis and management of foot deformities like flatfoot or cavus foot deformity, as well as in the planning and evaluation of surgical interventions. For instance, it has been demonstrated that specific foot dimensions, such as arch height or foot width, are associated with the development of foot conditions like plantar fasciitis [21,22,23]. In sports medicine, foot anthropometry has contributed to the understanding of biomechanical factors associated with foot and lower limb injuries in athletes [21]. It has assisted in injury prevention strategies and in identifying potential risk factors for lower limb injuries, facilitating the implementation of preventive measures [22]. Furthermore, podiatrists and manufacturers of footwear and orthotic devices can utilize foot anthropometry data to optimize product design tailored to individual patient needs, enhancing comfort and performance [9,10]. Additionally, the use of anthropometric measures in the design of orthotic devices has been shown to improve gait patterns and alleviate foot pain in patients with conditions like knee osteoarthritis [24].

In this view, obtaining reliable foot and ankle measurements is crucial for clinicians, shoe manufacturers, or even anthropologists [5,6,7]. The conventional tape measure can be unreliable [1,2] because the experience of clinicians significantly impacts the accuracy and the reproducibility of this method [3]. Another widely used conventional technique, the ink print of the foot arch, does not allow vertical foot measurements such as the navicular height, which are important factors for both clinical decision-making and shoe manufacturers [4]. Considering the increasing need for more accurate and reproducible measures of a large number of different anthropometric parameters, several foot and ankle scanners have become available since the 1990s for the clothing and shoe industries. The aim of these devices has been to update the anthropometric foot and ankle data of the general population and to adapt the design of the products as necessary [8,9,10]. This new technology has been associated with more reliable anthropometric measurements than conventional methods [4]. Key features of this new technology included swift acquisition, high accuracy, user-friendly interfaces, and relatively low costs [11]. Scanner analysis has allowed the characterization of the shape and size of large series of feet, whether in adults [12] or in children [13], and played an important role in confirming gender differences in foot sizes [14]. In addition to the accurate mapping of the skin surface, when used in combination with specific weight-bearing surfaces, they have allowed the characterization of the plantar support of the foot. For the first time, a large-scale, low-cost, computerized production of plantar orthoses was possible [15,24]. Nowadays, these scanners are routinely used for manufacturing custom-made orthopedic shoes and orthoses that adapt to patient-specific foot deformities [25].

Among available scanners, laser-based technologies remain the gold standard, while optical scanners represent a lower-cost alternative [24]. Powerful imaging software is capable of processing a large number of anthropometric points acquired by the scanner, without the need for external markers manually positioned on the foot and ankle. This feature allows the acquisition of anthropometric data in a semi- or fully automated fashion [26,27,28]. The standardization of an anthropometric model of skin markers was issued in 2018 (ISO 20685-1:2018), replacing the use of geometric points created by computerized systems of orthonormal coordinates. Therefore, most modern scanners use relatively uniform cutaneous landmarks, with a reported average accuracy of one millimeter or less [29].

To date, no study has comprehensively investigated the reproducibility of a large set of anthropometric measurements obtained with a 3D scanner.

The objective of this study was to validate an optical scanner for the reproducibility of length, width, and circumference measurements for thirty-nine comprehensive, key anthropometric points of the foot and ankle. Our initial hypothesis was that our laser scanner had high reproducibility levels, allowing its use in the clinical follow-up of patients.

## 2. Materials and Methods

We evaluated the reproducibility of ankle and foot anthropometric measurements performed using a laser scanner with an automated interface. Measurements were carried out in a series of 20 healthy subjects who volunteered to participate, for a total of 40 feet and ankles. Written informed consent was obtained from all participants. This non-interventional study was approved by the institutional review board for Paris East hospitals on 29 September 2021 (decisions Si-RIPH2G: 21.01741.000023 and N°RCB 2021-A01802.39).

The device tested was the optical/laser UPOD-S 3D Laser Full-Foot Scanner (manufactured in East Lake, Wuhan City, Hubei Province, China 430075), which is non-irradiating and transportable (13 kg, 27 × 52 × 22 cm). It is currently used for orthoses and orthopedic shoes manufacturing and allows the generation of a 3D model of the foot and ankle, up to a maximum height of 11.5 cm from the plantar support. The manufacturer claims 0.5 mm accuracy and a full scan time ranging between 5 and 15 s. The scanner comes with a software suite (UPOD 3D Full Foot Scan) allowing the identification of several anthropometric measurements in a fully automated way (Table 1 and Figure 1). After scanning, the software exports a PDF document with thirty-nine measurements: length (Table 2a and Figure 2a), width (Table 2b and Figure 2b), height (Table 2c), and ankle/foot circumferences (Table 2d and Figure 2c). Subjects were positioned upright, in bipedal support (one foot inside the scanner, the other on a footrest at the same height) with each foot bearing 50% support (Figure 3). Each foot was scanned individually. In this study, both feet were scanned three times during a 15 min session for each patient. An evaluation of the performance of this scanner in comparison to conventional methods has been previously conducted by Lee et al. [4], which concluded that 3D scanner analysis of foot anthropometrics is recommended, given its excellent precision and accuracy. The anthropometric data that we measured were similar to those evaluated in a study by Witana et al. [30].

All statistical analyses were carried out by an independent statistician. The centimetric measurements (lengths, widths, and circumferences) for each foot were performed three times per subject. First, the measurement variance for each foot was calculated, allowing us to obtain intra-subject variability that is only related to the specific measuring instrument. Then, average variance of measurements was calculated as follows: “average of the variances” = (variance for foot 1 + variance for foot 2 + variance for foot 3 +... + variance for foot 40)/40.

For each anthropometric measurement, the mean differences between measurements 1 and 2, 1 and 3, and 2 and 3 were then calculated to provide the “mean difference” of overall measurements (average difference measurements 1 and 2) + (average difference measurements 1 and 3) + (mean difference measure 2 and 3)/3. This measurement was an indicator of the magnitude, in millimeters, of the error made by the scanner while performing the three measurements and for similar anthropometric parameters. We also correlated the mean of the variances to the millimetric mean for each anthropometric parameter [31]. The millimetric mean of each item was defined as the mean of all measurements made [(mean of the 3 foot 1 measurements) + (mean of the 3 foot 2 measurements) + ... + (mean of the 3 foot 40 measurements)]/40. The “variance percentage” formula was = mean/mean variance of measurements. This variance percentage was an indicator of measurement dispersion related to the absolute value of the measurement itself.

The reproducibility of measures was considered excellent for a variance <1, good for a variance ≥1 and <5, and low for a variance ≥5.

Two datasets were analyzed. The first dataset included anthropometric parameters measured between two points, grouping the measurements of length, width, and height. The second dataset included circumferences, resulting from the measurement of perimeters drawn between different anthropometric points. For the measurement of anthropometric parameters (in mm), data on length, width, and height were considered together when their value was below 200 mm. Similarly, data on circumferences were considered together when their value was between 240 and 330 mm.

## 3. Results

Patient characteristics are shown in Table 3. Fifteen patients were female and five were male. Mean age was 35.62 +/− 9.54 years, range 9–75. Patient-reported shoe size ranged from 32 to 45.

Overall, the mean variance was 1.4 ± 2 (range 0.1 to 8), and the mean variance percentage was 1.3 ± 1.7 (range 0.1 to 8.3). The mean difference of measurements in millimeters was 0.3 ± 0.4 (range 0.04 to 1.9), with a mean absolute measurement error <1 mm.

Data obtained on length and width allowed us to identify measurements with very high reproducibility, notably “Foot”, “Arch”, “Medial Malleolus”, “Lateral Malleolus”, “ Fibular instep”, “1 Met to Pternion”, “5 Met to Pternion”, “HC to Pternion”, “Lat Arch to Pternion”, “Med Arch to Pternion”, “Waist Point to Pternion”, “Fore foot”, “Heel”, “Bimalleolar”, “Mid-Foot”, “1–5 Met”, “Toe 5 Outside”, “Metatarsale Tibiale”, “Metatarsale Fibulare”, and “Waist Point Outside”. Only three height parameters showed high reproducibility: “Ball girth”, “Instep”, “medial malleolus”, and “Mid-foot”. For all these measurements, the mean variance was <1, and the mean measurement difference was below 1 mm. The maximal variance percentage was 1.3%. Table 4 (green boxes).

For eight length, width and height measurements, the reproducibility was slightly lower: “Toe 1 Med to Pternion” (length), “Toe 1 inside” (width), and “Toe 1”, “Toe 5, “Navicular”, “Sphyrion, “Medial malleolus”, and “Lateral malleolus” (height). These measurements had a variance ranging from 1 to 5, with a mean difference lower than 1 mm and a variance percentage <6%. Table 4 (yellow boxes for length and height).

Lower reproducibility was found for two length and height measurements: “Toe5 lat to pternion” and “Sphyrion fibulare”, respectively. These measurements had a variance >5, mean difference lower than 1 mm, and a maximum variance percentage reaching 8.3%. Table 4 (red boxes for length and height).

Measurements of forefoot and midfoot circumferences “Ball”, “Instep”, “Short heel”, and “Waist” had a variance <2.5, with variance percentages <1%. The mean difference between the three sets of measurements did not exceed 1 mm. Table 4 (yellow boxes for circumference measurements). However, the hindfoot circumferences “Long heel” and “Ankle” had the highest variances in our series (7 and 5.9, respectively), with variance percentages >2%. The mean difference between the two measurements was between 1 and 2 mm. Table 4 (red boxes for circumference measurements).

## 4. Discussion

We comprehensively evaluated thirty-nine different foot and ankle anthropometrics, which represents a significant improvement in comparison to previous research [4,28,32,33]. This study, based on measurements of length, width, height, and circumference, confirms the high reproducibility of most of the foot and ankle anthropometrics obtained with the UPOD-S 3D Laser Full-Foot Scanner. Lee et al. [4], evaluating the same device, found superior performance in contrast to conventional methods such as digital caliper, digital footprint, and ink footprint, with excellent intra-class correlation coefficients (ICCs). However, their study was small and included only six anthropometric measurements. In the present study, the variance was low for every measurement, the mean difference was negligible, and the differences between three sets of measurements for each foot and ankle was <1 mm. Therefore, our findings validate this scanner as it allows a comprehensive and reproducible evaluation of foot and ankle anthropometrics.

It important to notice that the reproducibility of certain anthropometric data was slightly lower for a set of measurements:-The first metatarsal and fifth metatarsal ground support points (“toe 1 inside”, “toe 5 outside”, “Toe 1 Med to Pternion”, and “Toe 5 Lat to Pternion”) had, overall, the highest variances for length and width measurements, with a variance percentage ranging from 2% to 4%. During the data acquisition, the load appeared to be evenly distributed on the two feet, thanks to a platform that supported the contralateral foot at the same height as the foot undergoing the examination. The slightly higher variance could have resulted from asymmetric weight distribution applied by the subject on both feet.-The set of malleolar height points (“sphyrion”, “sphyrion fibulare”, “medial malleolus”, and “lateral malleolus”) also showed higher variance percentages, ranging from 1% to 8%. A possible explanation for a higher variance percentage is that the bony prominences of the malleolar points mentioned above appear more salient when scanned from the back. On the other hand, when they are scanned from the front, the transition between the proximal and distal areas of the anterior aspect of the foot shows a more arcuate shape, with a gentle slope, causing detection issues for the scanner, related to a mismatch of the measurements from the front and the back.-The measurements of the circumference had the highest variances compared to length, width, and height, but they also had the highest absolute values. The variance percentage reached 2.3% for those measurements with absolute values greater than 250 mm. The less reproducible circumference measurement was “long heel”; we suppose that the rotation of the leg in the scanner could have slightly affected the angle measurements at points of anatomical continuity between the foot and ankle, the so-called “junction points”. On the other hand, the reproducibility of circumferences outside the junction points was excellent, with a mean absolute error of around 1 mm.

Overall, the UPOD-S 3D Laser Full-Foot Scanner showed an excellent reproducibility for the vast majority of foot and ankle anthropometrics, with a mean absolute error lower than 1 mm. Malleolar height measurements were less repetitive, due to anatomical differences between the front and the back of the ankle, as explained above. The first and fifth metatarsal ground support points similarly showed a lower reproducibility in comparison to the other highly reproducible measures. Lastovicka et al. [32] also evaluated this scanner. They compared scanner measurements with manual tape measure but did not analyze the reproducibility of anthropometric data. They found excellent reliability (ICC > 0.98) when measuring lengths and widths (foot length and width, width of the median isthmus), but lower reliability when measuring heights, notably the height of the medial arch (CC = 0.62). They found excellent correlation with data obtained with manual tape measure (correlation ratio > 0.92), but also established that the correlation was lower when measuring the height of the medial arch. They further highlighted that measurement overestimation or underestimation could have been caused by clinicians, with the level of experience influencing the accuracy and the reproducibility of the manual methods. Variability could indeed result from the compression of soft tissues, from the positioning of the tape on salient skin features of the medial arch, or from changes in the ground support distribution among subjects [24]. Several previous studies have shown that 3D foot/ankle scanners have many advantages in comparison to conventional manual methods [4,23,28,34]. However, most of these studies have compared scanner measurements with data obtained via manual tape measure. To the best of our knowledge, the reproducibility of scanner measurements has rarely been investigated. As for height measurements, notably medial arch height, previous surveys have consistently found lower reliability and reproducibility. Using a Kinect sensor-optical system, Rogati et al. [28] found reproducibility scores of 0.99 and 0.93 when measuring foot length and forefoot width, while it was only 0.80 for height and 0.82 for internal arch width. Lee et al. [4] compared data obtained with various measurement methods (3D foot scan, manual tape measure, ink footprints, and baropodometric sensors) and found that the most reliable results were obtained with the scanner, with ICCs ranging from 0.95 to 0.98, and an overall accuracy around 1 mm.

To date, consistent with our results, the measurements of malleolar anthropometrics have been associated with challenges of reproducibility. De Mits et al. [34], using the IFOOT 3D digitizer, compared manual tape measures and data obtained with computerized landmarks and found that, for both methods, malleolar anthropometric points had the lowest correlation ratios (ICC, 0.80 to 0.86).

Surveys on circumference measurements are scarce, because they are known to be much less accurate in comparison to width and heights. De Mits et al. [34] obtained excellent ICCs with the INFOOT 3D digitizer, even for circumferences (>0.92), but with a standard deviation that increased significantly when measuring lengths (0.31 and 3.51 mm), heights (0.74 and 5.58 mm), and circumferences (0.75 and 5.9 mm). Zhao et al. [9] found errors <5 mm, provided salient skin features were mapped with high accuracy. They also found a systematic 4 mm underestimation of data obtained via manual tape measure. Witana et al. [30] found an excellent correlation rate for 10 of 18 measurement points, but differences were noted for the remaining 8 points. They thus identified “common-core” measurements (with high accuracy in both computerized and manual techniques) in contrast with “other” measurements, in which the mapping of salient skin features was different in computerized and manual techniques (heel-to-fifth toe length, heel width, bimalleolar width, mid-foot width, height at 50% of the foot length, long heel girth, ankle girth, and waist girth). Seven of these eight “other” measurements included height circumference measurements.

Although the ISO 20,685 standard issued in 2018 has allowed the standardization of scanner specifications, there remain several differences in how certain foot measurements are defined by authors [30], notably when it comes to the ground reference axis for projected lengths. It is likely that, in the future, progress in image processing software suites will increase the reproducibility and reliability of data. Finally, as all available scanners operate with an optical system, artifacts induced by natural light can also cause measurement inaccuracy. The scanner used in our study is an improved version of a previous device, and includes a unique feature (i.e., panels protecting the foot up to the instep), thereby shielding it from natural light and probably allowing more reliable acquisition. All round, our scanner’s specifications make it extremely reliable, with only a handful of less accurate anthropometric measurements. The precision and accessibility of foot anthropometry are expected to further improve, enhancing its clinical utility in the years to come. Given the growing interest in the fields of imaging and predictive outcomes, and the increasing affordability of artificial intelligence and computational power, this technology will be used in the future to establish machine learning models.

There are some limitations to this study, concerning in general the accuracy and reproducibility of anthropometric measurements carried out with optical scanners. First, our results refer to a series of healthy subjects, while severe foot and ankle deformity is associated with alteration of the normal surface anatomy and could confound scanner measurements based on automated computerized interfaces. This limitation is shared with previous studies that have validated data obtained in a healthy population with minimal deformation of salient cutaneous points. To the best of our knowledge, no study involving foot/ankle scanner anthropometric measurements has yet reported reproducibility scores for severe deformity. Second, foot and ankle edema and obesity could also lower the reproducibility of the anthropometric measurement by altering the surface anatomy. The amount of subcutaneous fat or edema was not measured in this study and is thought to influence circumferences and heights more than lengths, for reasons of volume distribution in the foot and ankle. On the other hand, we evaluated a large set of anthropometrics, and measurements were repeated three times for each subject. Given the fast acquisition time of the scanner, foot and ankle volume fluctuations during the examination would have hardly played a role in lowering the reproducibility of the measurements.

Third, a comparison of patients in the supine and sitting position would have allowed us to evaluate whether the plantar pressure caused by the standing position is associated with modifications of the anthropometric measurements. However, weight-bearing CT scan is considered the gold standard for preoperative planning in the field of foot and ankle surgery. Fourth, we did not evaluate the influence of gender with respect to anthropometric measurements; nevertheless, a subgroup analysis was not possible given the small number of male patients included in the present study.

## 5. Conclusions

The UPOD-S 3D Laser Full-Foot Scanner achieved good reproducibility scores when measuring a large set of foot and ankle anthropometrics, with an overall mean measurement error <1 mm. Lower reproducibility was observed for few anthropometric measurements: height of malleolar points, first and fifth metatarsal plantar supports, and hindfoot circumference. Considering its speed of execution, user-friendly features, and high automation, this device is an excellent tool for the manufacturing industry, and for clinical and research purposes. A patent derived from the present study might promote the application of this technology in clinical practice and improve preoperative decision-making and patient follow-up.

## Figures and Tables

**Figure 1 bioengineering-10-00968-f001:**
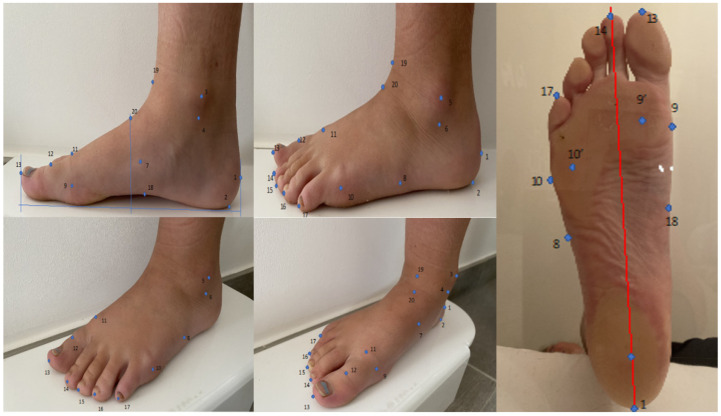
Representation of the surface anatomy of the foot and ankle with the identification of different anthropometric points which are described in Table 1.

**Figure 2 bioengineering-10-00968-f002:**
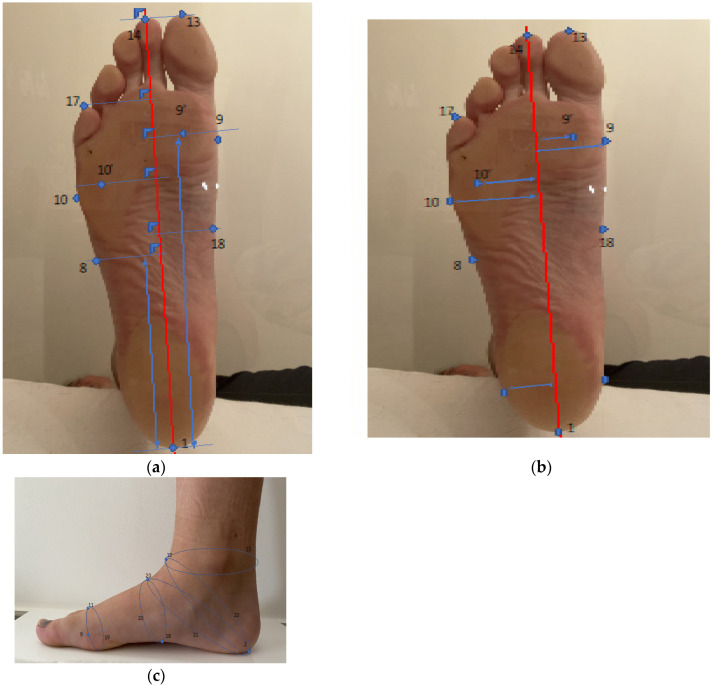
(**a**) Lengths were measured from the pternion along the axis of the foot (red line between pternion and toe 2). These measurements are distances projected on the ground and measured from the longitudinal axis of the foot (axis between heel support and O2 indicated in red on the figure). Instep is located at 50% of foot length from pternion. (**b**) Widths were measured from the point perpendicular to the foot axis (red line between pternion and toe 2). (**c**) Measurement of circumferences.

**Figure 3 bioengineering-10-00968-f003:**
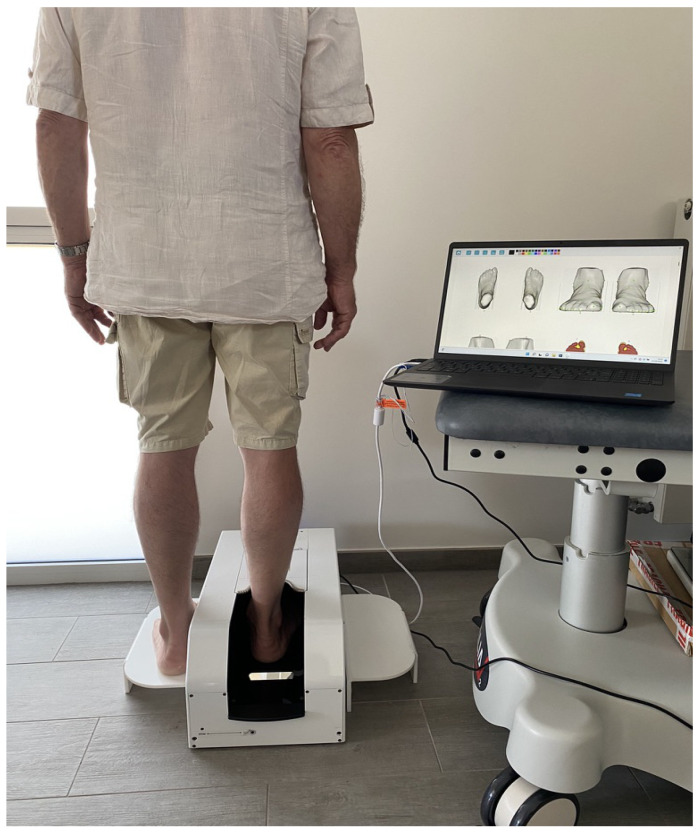
Subjects were positioned standing on both feet: one foot in the scanner, and the other foot on a footrest at the same height (courtesy of Beldame J et al., Assessment of the Efficiency of Measuring Foot and Ankle Edema with a 3D Portable Scanner. *Bioengineering*, 2023).

**Table 1 bioengineering-10-00968-t001:** The identification of twenty different anatomical points on the surface of the foot and ankle allowed the evaluation of thirty-nine anthropometric measurements.

Number	Name of Point
1	Pternion
2	Landing points
3	The most medial point of medial malleolus
4	Sphyrion
5	The most lateral point of lateral malleolus
6	Sphyrion fibulare
7	Navicular (the most medial point of navicular landmark)
8	Tuberosity of 5th metatarsal
9	Metatarsal tibiale
10	Metatarsal fibulare
11	Highest point of 1st metatarsal head
12	Toe 1 joint
13	Tip of 1 toe
14	Tip of 2 toe
15	Tip of 3 toe
16	Tip of 4 toe
17	Tip of 5 toe
18	Highest point of medial arch
9′	Ground support of M1
10′	Ground support of M5
19	Junction point
20	Highest point of Instep without the case of 50% of foot length

**Table 2 bioengineering-10-00968-t002:** (a) Definitions of foot dimensions—lengths. « Axis » is defined as the axis between the pternion point and the axis of the second toe (point 14). (b) Definitions of foot dimensions—width. « Axis » is the axis between pternion point and the axis of the second toe (point 14). (c) Definitions of foot dimensions—heights. (d) Definitions of foot dimensions—circumferences.

(a)			
**Length in mm**	Foot length		Distance along the axis from pternion to the tip of the longest toe
	Arch length		Distance along the axis from pternion to the most medially prominent point on the first metatarsal head
	Medial malleolus		Distance along the axis of the most medial point of medial malleolus
	Lateral malleolus		Distance along the axis of the most lateral point of lateral malleolus
	Fibulare instep		Distance along the axis of the most lateral point of instep
	1met to pternion		Distance from contact point of M1 to pternion
	5met to pternion		Distance from contact point of M5 to pternion
	HC to pternion		(Horizontal) distance between center point of lateral and medial malleolus to pternion
	Lat arch to pternion		Distance along the axis of the most lateral point of the arch
	Med arch to pternion		Distance perpendicular to axis of the most lateral point of the arch
	Toe 1 med to pternion		Distance from the most medial point of M1 to pternion
	Toe 5 lat to pternion		Distance from the most lateral point of M5 to pternion
	Waist point to pternion		Distance along the axis of the highest point mid foot, at 50% of foot length from pternion
(b)			
**Width in mm**	Forefoot width		Distance between horizontal breadth, across the foot axis in the region in front of the most laterally prominent point on the fifth metatarsal head
	Heel width		Breath of the heel, 40 mm forward of the pternion
	Bimalleolar		Distance between the most medially protruding point on the medial malleolus and the most laterally protruding point on the lateral malleolus, measured perpendicular to axis
	Width mid-foot		Maximum horizontal breath, across the foot perpendicular to axis, at 50% of foot length from the pternion
	1–5 toe met		Maximum horizontal breath across the foot, perpendicular to axis, passing by toe 1 inside and toe 5 outside
	Toe 1 inside		Toe 1 (big toe) contact point
	Toe 5 outside		Toe 5 (little toe) contact point
	Metatarsale tibiale		Width of the most medial point of M1 to axis
	Metatarsale fibulare		Width of the most lateral point of M5 to axis
	Waist point outside		The most lateral point relief, across the foot perpendicular to axis, at 50% of foot length from the pternion
(c)			
**Height * in mm**		Ball girth	Height of the highest point of ball girth circumference
		Instep	Height of the highest point at the level of 50% of foot length
		Toe 1	Height of the highest point of M1
		Toe 5	Height of the highest point of M5
		Navicular	Height of navicular point
		Sphyrion	Height of sphyrion point
		Lateral malleolus	Vertical distance from the floor to the most prominent point on the lateral malleolus
		Medial malleolus	Vertical distance from the floor to the most prominent point on the medial malleolus
		Mid-foot	Maximum height of the vertical cross-section at 50% of foot length from the pternion
(d)			
**Girth in mm**		Metatarsal girth	Circumference of foot, measured with a tape touching the medial margin of the head of the first metatarsal bone, top of the first metatarsal bone and the lateral margin of the head of the fifth metatarsal bone
		Instep girth	Circumference at the level of midfoot, at 50% of foot length
		Long heel girth	Girth from instep point around back heel point
		Short heel girth	Maximum girth around back heel point and dorsal foot surface
		Ankle girth	Horizontal girth at the foot and leg intersection
		Waist	Smallest girth over middle cuneiform prominence

* These measurements correspond to the height of anthropometric points from the ground.

**Table 3 bioengineering-10-00968-t003:** Demographic characteristics of the population.

Population	*n* = 20
Gender (Male/female)	5/15
Age (years) average +/− standard deviation	35.62 +/− 9.54
Minimum age (years)	9
Maximum age (years)	75
European shoe size (average +/− standard deviation) Range	38.17 +/− 3.23 32–45

**Table 4 bioengineering-10-00968-t004:** Overview of the measurements of foot and ankle anthropometrics. The color of boxes refers to higher reproducibility of the measurements (green), or to lower reproducibility (yellow and red).

		Mean Variance *	Mean Difference (in mm) **	Mean Measurement (in mm) ***	Mean % of Variance ****
Length measurements	Foot	0.537	0.126	250.39	0.21
	Arch	0.266	0.07	180.29	0.15
	Medial Malleolus	0.619	0.151	60.67	1.02
	Lateral Malleolus	0.693	0.201	54.02	1.29
	Fibulare instep	0.206	0.083	157.73	0.13
	1 Met to Pternion	0.477	0.134	180.29	0.27
	5 Met to Pternion	0.376	0.171	157.73	0.24
	HC to Pternion	0.144	0.138	32.51	0.44
	Lat Arch to Pternion	0.236	0.062	95.51	0.25
	Med Arch to Pternion	0.638	0.137	106.72	0.6
	Toe 1 Med to Pternion	3.538	0.791	222.61	1.59
	Toe 5 Lat to Pternion	8.006	0.283	189.95	4.22
	Waist Point to Pternion	0.095	0.05	105.91	0.09
Width measurements	Fore foot	0.222	0.277	98.35	0.23
	Heel	0.159	0.104	61.18	0.26
	Bimalleolar	0.242	0.111	71.51	0.34
	Mid-Foot	0.27	0.171	85.10	0.32
	1–5 Met	0.0646	0.316	65.71	0.98
	Toe 1 inside	1.012	0.127	45.77	2.13
	Toe 5 Outside	0.598	0.081	44.57	1.34
	Metatarsale Tibiale	0.086	0.16	47.78	0.18
	Metatarsale Fibulare	0.073	0.109	47.93	0.15
	Waist Point Outside	0.128	0.04	41.13	0.31
Height measurements	Ball girth	0.286	0.116	41.99	0.68
	Instep	0.374	0.183	70.15	0.53
	Toe 1	0.535	0.141	25.90	2.07
	Toe 5	0.52	0.14	21.21	2.45
	Navicular	0.668	0.194	41.56	1.62
	Sphyrion Fibulare	5.068	0.633	62.01	8.28
	Sphyrion	1.727	0.562	71.29	2.42
	Lateral Malleolus	4.225	0.619	72.29	5.89
	Medial Malleolus	1.156	0.416	87.29	1.33
	Mid-foot	0.368	0.15	70.61	0.52
Girth Measurements	Ball	1.514	0.256	243.10	0.62
	Instep	1.592	0.612	245.42	0.65
	Short Heel	2.445	1.013	342.38	0.71
	Long Heel	7.059	1.955	330.14	2.14
	Ankle	5.941	1.102	253.60	2.34
	Waist	1.485	0.223	245.41	0.6

* For each foot and ankle, measurement variance was calculated, allowing us to obtain intra-subject variability (solely due to the measuring device). Then, the mean variance of measurements was calculated as follows: “mean variance” = (variance for foot 1 + variance for foot 2 + variance for foot 3 +... + variance for foot 40)/40. ** The mean differences between measurements 1 and 2, 1 and 3 and 2 and 3 were then calculated, allowing us to obtain the “mean difference” of measurements = [(mean difference between measurement 1 and 2) + (mean difference between measurement 1 and 3) + (mean difference between measurement 2 and 3)]/3. *** The mean measurement for each item was the mean of all measurements = (mean of the three measurements for foot 1 + mean of the three measurements for foot 2 + mean of the three measurements for foot 3)/40. **** The mean percentage of variance = (mean variance/mean of measurements) × 100.

## Data Availability

Research data is not publicly archived and is available on request from the corresponding author.

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
