# Peer review of "Validation of an Automated Optical Scanner for a Comprehensive Anthropometric Analysis of the Foot and Ankle"

_bioengineering, 2023, doi:10.3390/bioengineering10080968_

Round 1

Reviewer 1 Report

The paper submitted for review deals with validating an automated optical scanner, which could provide a comprehensive anthropometric analysis of the foot and ankle.

However, I have some doubts or suggestions which will improve the quality of your article.

Line 114: Besides positioning standing on the scanner, I suggest the author add the test in which subjects sit in the chair with their feet on the scanner. This test could analyze the anthropometric indicators at lower pressures (even tend to be none). During the surgery or foot orthopedics et al., the foot is suspended.

Line 297: The reason for the lowest correlation ratios of malleolar anthropometric points may be the methods of measurement. It was different in plantar structure between standing position and sitting position. That’s why I suggest the author added the data on the sitting position.

Line 157: To avoid misunderstandings, I suggest the author add the definition and common analysis method of reproducibility, which should be supported by reference.

Line 243: The explanation was unreasonable. If the weight distribution of subjects’ feet were asymmetric, why the other indicators (associated with the weight distribution) were not affected?

The formatting of tables needs to be revised to the three-line table.

Author Response

August 3rd, 2023

Paris, France

Manuscript ID bioengineering 2527477

Dear Editors,

Dear Reviewers,

Thank you for reviewing our paper “ Validation of an automated optical scanner for a comprehensive anthropometric analysis of the foot and ankle“.

Submitted for possible publication in Bioengineering.

We have reviewed the comments of the reviewers and have replied to all queries.

We found all the comments extremely helpful, and believe that our revised represents a significant improvement over our initial submission.

Yours sincerely,

The authors of the article bioengineering 2527477.

All modifications made as per reviewers suggestions have been highlighted in red in the Manuscript.

Reviewer #1

The paper submitted for review deals with validating an automated optical scanner, which could provide a comprehensive anthropometric analysis of the foot and ankle. However, I have some doubts or suggestions which will improve the quality of your article. 

Reply: We thank reviewer #1 for her/his positive comment.

—-

Line 114: Besides positioning standing on the scanner, I suggest the author add the test in which subjects sit in the chair with their feet on the scanner. This test could analyze the anthropometric indicators at lower pressures (even tend to be none). During the surgery or foot orthopedics et al., the foot is suspended. 

This is an interesting observation. Previous research has focused on supine or standing position for the acquisition of CT scan foot anthropometrics. The present study was entirely conducted in the standing position, for this reason a comparison of anthropometrics in the standing and sitting position was not possible. This limitation was added in the discussion. Please see the revised text - Lines 349 - 352.  

“Third, a comparison of patients in the supine and sitting position would have been useful to evaluate whether the plantar pressure caused by the standing position is associated with modifications of the anthropometric measurements. However, weight bearing CT scan is considered the gold standard for preoperative planning in the field of foot and ankle surgery.”

____

Line 297: The reason for the lowest correlation ratios of malleolar anthropometric points may be the methods of measurement. It was different in plantar structure between standing position and sitting position. That’s why I suggest the author added the data on the sitting position.

The present study was entirely conducted in the standing position, for this reason a comparison of anthropometrics in the standing and sitting position was not possible. This limitation was added in the discussion. Please see the revised text as above - Lines 348 - 352.

____

Line 157: To avoid misunderstandings, I suggest the author add the definition and common analysis method of reproducibility, which should be supported by reference.

The reference “31” was added. 31. Flannelly KJ, Jankowski KR, Flannelly LT. Measures of variability in chaplaincy, health care, and related research. J Health Care Chaplain. 2015;21(3):122-130. doi:10.1080/08854726.2015.1054671

—-

Line 243: The explanation was unreasonable. If the weight distribution of subjects’ feet were asymmetric, why the other indicators (associated with the weight distribution) were not affected?

We agree that this consideration is unreasonable according to the evidence of our results. The text was deleted, please see the revised text, deleted lines 244-246.

—-

The formatting of tables needs to be revised to the three-line table.

Table numbers were modified including consequent subfigure numbers (Figure 1a, 1b, 1c…). We believe that the present formatting of tables is necessary to improve the reader's understanding of the article, given the large number of variables evaluated. On the other hand, we agree that the tables can be converted to a three-line format in the final editing process if necessary.

Reviewer 2 Report

In the present work, The Authors investigated the reproducibility of a large set of anthropometric measurements, taken from the foot and ankle, obtained with a 3D scanner. In a previous work from the same Authors, volumetric measurements of foot and ankle were assessed using the same scanner and compared to gold standard. Now they focus on a larger set of measurements. The approach is rigorous and methodologically correct. However, some points need to be fixed.

  1. Please, rearrange enumeration of tables and figures according to the Journal’s guidelines. If the maximum number of tables is exceeded, consider moving to the Supplementary section (eg. Tables 1, 2X). Accordingly, group Figures 1xy into a single figure with subfigures numbered as 1a, 1b, 1c, etc…

  2. When defining the mean percentage of variance as mean variance / mean of measurements (reported in Table 4), it is not clear whether a 100 factor was applied when calculating percentage.

  3. It could be interesting to evaluate the effect of sex difference on measurements, even though the sample is unbalanced, with predominant female gender.

  4. In Table 3, shoe size reporting (mean and std.dev.) may be misleading, please adjust horizontal alignment.

  5. Figure 2S has been already published in a previous work by the Authors [17]. This should be clearly stated.

  6. Please check bibliography enumeration: entries from 16 to 22 have double enumeration.

Only a few typos need to be fixed

Author Response

August 3rd, 2023

Paris, France

Manuscript ID bioengineering 2527477

Dear Editors,

Dear Reviewers,

Thank you for reviewing our paper “ Validation of an automated optical scanner for a comprehensive anthropometric analysis of the foot and ankle“.

Submitted for possible publication in Bioengineering.

We have reviewed the comments of the reviewers and have replied to all queries.

We found all the comments extremely helpful, and believe that our revised represents a significant improvement over our initial submission.

Yours sincerely,

The authors of the article bioengineering 2527477.

All modifications made as per reviewers suggestions have been highlighted in red in the Manuscript.

______________________________________________

Reviewer #2

In the present work, The Authors investigated the reproducibility of a large set of anthropometric measurements, taken from the foot and ankle, obtained with a 3D scanner. In a previous work from the same Authors, volumetric measurements of foot and ankle were assessed using the same scanner and compared to gold standard. Now they focus on a larger set of measurements. The approach is rigorous and methodologically correct. However, some points need to be fixed.

Reply: We thank reviewer #2 for her/his positive comment.

  1. Please, rearrange enumeration of tables and figures according to the Journal’s guidelines. If the maximum number of tables is exceeded, consider moving to the Supplementary section (eg. Tables 1, 2X). Accordingly, group Figures 1xy into a single figure with subfigures numbered as 1a, 1b, 1c, etc…

Table numbers were modified including consequent alphabetic subfigure numbers (Figure 1a, 1b, 1c…). We reviewed again Bioengineering guidelines for authors and we did not find any statement addressing a limit for the number of tables and figures. Given the large set of data presented, we think that the current format (tables and figures grouped according to the type of anthropometrics), and the number of tables and figures, is necessary for the reader's understanding of the article. If the number of tables and figures in the main text needs to be reduced, Figure 2S and Table 4 can be moved to the Supplementary material.

  1. When defining the mean percentage of variance as mean variance / mean of measurements (reported in Table 4), it is not clear whether a 100 factor was applied when calculating percentage.

Thank you to make us notice this detail. The formula in the description of Table 4 was defined as follows . **** The mean percentage of variance = (mean variance / mean of measurements) x 100.

  1. It could be interesting to evaluate the effect of sex difference on measurements, even though the sample is unbalanced, with predominant female gender.

Given the small number of males (n=5) compared to females (n=15)  a subgroup analysis was not possible. Further studies could evaluate potential differences in foot and ankle anthropometrics according to patients sex. This observation was added in the limitation section. Please see the revised text, Lines 353-355.

Fourth, we did not evaluate the influence of gender with respect to anthropometric measurements; nevertheless, a subgroup analysis was not possible given the small number of male patients included.

  1. In Table 3, shoe size reporting (mean and std.dev.) may be misleading, please adjust horizontal alignment.

Table 3 legibility was improved.

  1. Figure 2S has been already published in a previous work by the Authors [17]. This should be clearly stated.

The reference was added in the figure description of Figure 2S (now Figure 2).

Figure 2. Subjects were positioned standing on both feet: one foot in the scanner, and the other foot on a footrest at the same height. (Courtesy of Beldame J et al., Assessment of the Efficiency of Measuring Foot and Ankle Edema with a 3D Portable Scanner. Bioengineering, 2023).

  1. Please check bibliography enumeration: entries from 16 to 22 have double enumeration.

The typo error was corrected. Please see the reference list.